# FAZ Segmentation Quality Assessment in OCTA via Denoising Autoencoders and Segmentation Uncertainty Estimation

**Hana Jebril**[1]    HANA.JEBRIL@MEDUNIWIEN.AC.AT

**Guilherme Aresta**[1,2]    GUILHERME.ARESTA@MEDUNIWIEN.AC.AT

**Hrvoje Bogunović** [1,2]    HRVOJE.BOGUNOVIC@MEDUNIWIEN.AC.AT

[1] *Institute of Artificial Intelligence, Center for Medical Data Science, Medical University of Vienna, Austria*

[2] *Christian Doppler Lab for Artificial Intelligence in Retina, Center for Medical Data Science, Medical University of Vienna, Austria*

**Editors:** Accepted for publication at MIDL 2025

## Abstract

Accurate segmentation quality assessment is essential in medical imaging, particularly in preventing segmentation algorithms from failing silently. We propose a Segmentation Quality Assessment Framework method that estimates segmentation quality without relying on ground-truth labels. Our approach integrates learning-free uncertainty estimation with a Denoising Autoencoder (DAE) to generate pseudo-labels, extract key statistical features, and train a Random Forest Regressor (RDF) for quality prediction. Experimental results demonstrate that our method outperforms baseline approaches on three external datasets, showcasing its robustness to image domain shifts. our method enhances the scalability and generalizability of real-world medical imaging applications by leveraging segmentation models to handle cases where manual annotations are missing or infeasible.

**Keywords:** Quality Assessment, Uncertainty, Segmentation, Image Domain Shift.

## 1. Introduction

Optical Coherence Tomography Angiography (OCTA) imaging allows the segmentation of the retinal Foveal Avascular Zone (FAZ), a capillary-free region at the center of the fovea important for the diagnosis of both retinal and systemic diseases, but the task is very subjective, resulting in inconsistent FAZ measurements between observers (Fernández-Espinosa et al., 2023). Automated segmentation methods can improve segmentation reproducibility while removing the annotation burden from medical experts. Deep learning has become the standard for medical image segmentation. However, these methods often overconfidently fail due to changes in acquisition settings, image noise, or presence of artifacts. Identifying such failures is crucial to make these systems trustworthy, increasing their applicability in clinical practice (Aggarwal et al., 2011).

Namely, (Arega et al., 2023) proposed to estimate segmentation performance via a Random Forest Regressor (RDF) that receives as input the Dice similarity coefficient (DSC), Hausdorff distance (HD), and the means of uncertainty maps between multiple outputs obtained via Monte Carlo (MC) dropout. Other studies (Zhou et al., 2019; Zaman et al., 2023)

have introduced quality estimation models based on regression CNNs, where the segmentation map and the difference between the original and reconstructed images are directly fed into the network. However, these methods lack robustness to image domain shifts caused by variations in imaging devices or acquisition settings. Following this, we i) propose a segmentation quality assessment framework for FAZ segmentation in OCTA images, which integrates Denoising Autoencoder (DAE) (Vincent et al., 2010) with learning-free uncertainty estimation methods to predict segmentation errors while maintaining robustness to domain shifts in the images. ii) validate our approach across multiple OCTA datasets, demonstrating its robustness in estimating segmentation quality.

## 2. Method and Experiments

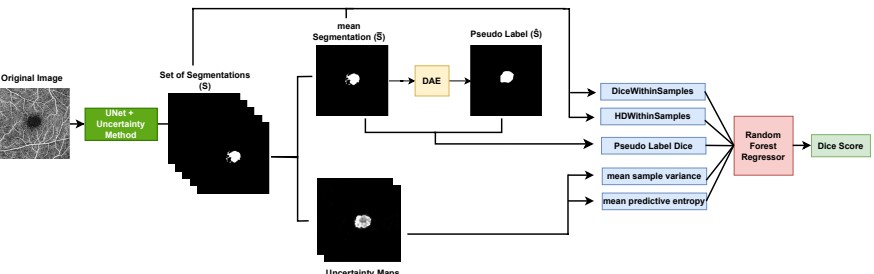

Figure 1: Proposed Segmentation Quality Assessment Framework

The proposed framework (Figure 1) first predicts a set of segmentations $S$ using any learning-free uncertainty estimation method (e.g. MC dropout, Deep Ensemble (DE) or Test Time Augmentation (TTA)). We then compute four uncertainty-related features from $S$: DSC, and HD between the mean prediction and each prediction map, mean sample variance, and mean sample entropy from their corresponding uncertainty maps. The mean segmentation $\overline{S}$ also serves as input to a DAE, which generates a pseudo-label segmentation $\hat{S}$. We then compute an additional feature, the DSC between $\overline{S}$ and $\hat{S}$. These 5 features are fed to a RDF, which predicts an estimation of the underlying ground-truth DSC.

**Model and training** The segmentation model is a Bayesian U-Net (Dechesne et al., 2021) trained with a weighted combination of cross-entropy and DSC losses. The DAE (Vincent et al., 2010) was trained to generate segmentations from degraded versions of manual references using as loss a combination of DSC and a boundary loss that measures the integral of the distance function over the boundary of the ground-truth segmentation.

The DAE input is created by randomly adding and removing large binary regions (Larrazabal et al., 2020). The RDF is trained to regress the DSC between $\overline{S}$ and the manual reference.

**Experiments** The OCTA-500 dataset (Li et al., 2019) was used for model development with a split of 70% training, 15% validation, and 15% testing. For external test set we use **FAZID** (Balaji et al., 2020) (304 scans), **OCTA-25K** (Wang et al., 2021) (1,012 scans), and **Rose** (Ma et al., 2021) (39 scans). We compare our method with the work of (Arega et al., 2023), which is based on MC dropout and extracts the same four features as ours, but without

Table 1: $R^2$ and Pearson Correlation Coefficient (PCC) values for the assessed methods. FAZID, OCTA-25K and ROSE are external datasets. LFUEM: learning-free uncertainty estimation method used to generate the segmentation variations.

| LFUEM | Approach | OCTA-500 | | FAZID | | OCTA-25K | | ROSE | |
|---|---|---|---|---|---|---|---|---|---|
| | | $R^2$ | PCC | $R^2$ | PCC | $R^2$ | PCC | $R^2$ | PCC |
| RECNet | (Zhou et al., 2019) | 0.195 | 0.576 | -6.687 | 0.094 | -1.174 | 0.585 | 0.587 | 0.843 |
| MC | (Arega et al., 2023) | 0.690 | 0.869 | 0.609 | 0.808 | -0.002 | 0.645 | 0.697 | 0.880 |
| MC | Ours | **0.727** | **0.879** | 0.638 | 0.828 | 0.179 | 0.678 | 0.752 | 0.903 |
| DE | (Arega et al., 2023) | 0.297 | 0.821 | 0.759 | 0.877 | 0.565 | 0.794 | 0.800 | 0.917 |
| DE | Ours | 0.614 | 0.834 | **0.831** | **0.916** | **0.704** | **0.865** | **0.872** | **0.935** |
| TTA | (Arega et al., 2023) | -0.077 | 0.568 | 0.550 | 0.765 | 0.583 | 0.800 | 0.596 | 0.790 |
| TTA | Ours | 0.260 | 0.667 | 0.795 | 0.908 | 0.620 | 0.823 | 0.748 | 0.875 |

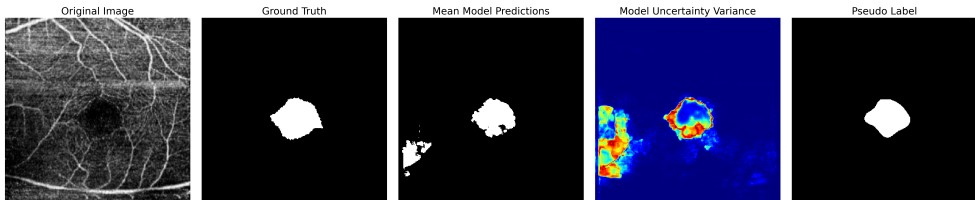

Figure 2: A qualitative example showing the mean model segmentation output, the segmentation uncertainty variance map, and the pseudo-label generated by the DAE.

incorporating the DAE-based feature. Additionally, we compare with RECNet (Zhou et al., 2019), which uses a regression AlexNet to predict the segmentation DSC from the input image, the inferred segmentation and a deep-learning based reconstruction of the input image using the inferred segmentation.

## 3. Results and Discussion

Table 1 shows $R^2$ and Pearson Correlation Coefficient (PCC) values between the true and predicted segmentation DSC scores for the assessed approached. A qualitative example is shown in Figure 2. Results indicate that the features from DAE reconstruction allow to improve the generalizability of the model. The proposed Segmentation Quality Assessment Framework is better at predicting segmentation quality than other approaches, especially for the DE method. Future work will extend the method to enable disease identification from segmented labels instead of raw imaging data.

## Acknowledgments

This research was funded in part, by the Austrian Science Fund (FWF) [10.55776/FG9].

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
