# OpenReview forum: "FAZ Segmentation Quality Assessment in OCTA via Denoising Autoencoders and Segmentation Uncertainty Estimation"
_MIDL.io/2025/Short_Papers — MIDL 2025 - Short Papers_

### Official Review · Reviewer_q5yP · 2025-04-28

**Rating:** 5
**Confidence:** 5

**Summary:**

The authors propose a segmentation quality control framework method to estimate segmentation quality without relying on ground-truth labels. This approach integrates learning-free uncertainty estimation with a Denoising Autoencoder to generate pseudo-labels, extract key statistical features, and train a Random Forest Regressor to predict the quality. This method is applied on OCTA images segmentation.

**Strengths:**

This paper has several strengths:
- new segmentation quality control
- integration of a Denoising Autoencoder (DAE)
- robustness to domain-shifts
- validation on multiple OCTA datasets

**Weaknesses:**

there is no specific weakness for this short paper.  More experiments with other imaging modalities and anomaly detection could be very interesting.

---

### Decision · Program_Chairs · 2025-05-01

Accept